# Preparation of Conductive and Corrosion Resistant Phosphate Conversion Coating on AZ91D Magnesium Alloy

Shudi Zhang [1,*], Yuheng Xu [1], Linkun Liu [1], Quanda Lei [1], Jialin Dong [1] and Tao Zhang [2,*]

1   School of Environmental and Chemical Engineering, Shenyang Ligong University, Shenyang 110159, China; lovexyh2023@163.com (Y.X.); 15670291571@163.com (L.L.); lqd317213741@163.com (Q.L.); 18845713710@163.com (J.D.)
2   Chinese Academy of Sciences (Shenyang) Metals Research, Shenyang 110016, China
*   Correspondence: zhangshuidi@163.com (S.Z.); zhangtao@mail.neu.edu.cn (T.Z.)

**Abstract:** The application of magnesium alloys in the 3C industry requires the coexistence of excellent corrosion resistance and good electrical conductivity. In this work, a conductive and corrosion-resistant phosphate conversion coating (PCC) on AZ91D magnesium alloy was investigated. The effects of strong oxidant ($KMnO_4$), additive ($Na_2MoO_4$), surface-active agent (OP-10) and their content in phosphating bath on PCCs were studied, and the mechanism of action of strong oxidant was analyzed. The results showed that the optimum content for $KmnO_4$, $Na_2MoO_4$ and OP-10 in phosphating bath was 3.0 g/L, 1.5 g/L and 1.0 g/L. The PCC formed at the phosphating bath at the optimum condition was completely covered, the coating on $\alpha$ phases had a bilayer structure and the $\beta$ phases were protruded. The electrical contact resistance (ECR) of the PCC was as low as 4.91 $\Omega$, the $E_{corr}$ positively shifted about 27 mV, and the $i_{corr}$ reduced significantly. The presence of $KMnO_4$ inhibited the formation of phosphate crystals and made the $\beta$ phases protrude from the surface to form conductive spots, which improved the conductivity of PCCs.

**Keywords:** phosphate conversion coating; magnesium alloy; strong oxidant; corrosion resistance; electrical conductivity





## 1. Introduction

Magnesium alloys are the lightest engineering materials [1], which have promising applications in aerospace, military defense, biomedicine, transportation and the 3C industry [2–10] due to their excellent properties such as low density, high mechanical strength, high thermal conductivity and good electromagnetic compatibility [11]. Unfortunately, the poor corrosion resistance of magnesium alloys is still the biggest challenge of expanding their application range [12]. Surface treatment technology is an effective means to improve the corrosion resistance of magnesium alloys. Thus, the surface treatment technologies for magnesium alloys including anodic oxidation, micro-arc oxidation, organic coating, electroplating, chemical plating and chemical conversion treatment have experienced rapid development in the past decades [13–17]. As the applications of magnesium alloys in the 3C industry gradually expanded, in order to meet the requirements such as electromagnetic compatibility and electrostatic release, it is necessary to develop a surface treatment technology which can provide a coating with both excellent corrosion resistance and good electrical conductivity.

Chemical conversion treatment has gained widespread interest due to its low cost and simplicity of operation [18]. However, the chemical conversion coatings of magnesium alloys are usually composed of oxides and insoluble salts, which leads to the lack of conductivity of the coating itself [19]. Considering that the electrical contact resistance (ECR) is determined by the electrical conductivity of the material [20], and the charge transfer resistance ($R_t$) related to corrosion resistance is the resistance of metal atoms to lose electrons and become cations during the electrochemical reaction [21], the two are

irrelevant. Therefore, it is feasible to prepare conversion coatings with high corrosion resistance and low electrical contact resistance.

A few studies on the chemical conversion coatings with high corrosion resistance and low electrical contact resistance have been reported. Jian et al. [22] developed a permanganate conversion coating with a thickness of only 230 nm on AZ31 magnesium alloy; this coating has sufficient conductivity to ensure the magnetic shielding performance of AZ31 due to its low thickness. Duan et al. [23] proposed a design idea of a chemical conversion coating with high corrosion resistance and low ECR on AZ91D magnesium alloy, which was to make $\beta$-$Mg_{17}Al_{12}$ phase protrude from the surface as a conductive spot by using the micro-galvanic effect between different phases and adding $VO_3^-$ as a strong oxidant to conversion solution to inhibit the deposition of coating on it. Zhou et al. [24] studied the phosphate conversion coatings (PCCs) with low electrical contact resistance on sand-cast and die-cast AZ91D magnesium alloy, and found that the PCCs of die-cast exhibit lower ECR and better corrosion resistance, which is attributed to the intensified micro-galvanic effect between different phases through the grain refinement of the die-cast alloy.

The main function of the phosphate conversion coatings is to improve the corrosion resistance of magnesium alloy. However, the current studies on conductive and corrosion-resistant PCCs mainly focus on how to improve their conductivity, and ignore the corrosion resistance. Therefore, the aim of this study is to prepare PCCs with low ECR and high corrosion resistance on AZ91D magnesium alloy. The pretreatment before phosphating was screened. The effects of strong oxidant ($KMnO_4$), additive ($Na_2MoO_4$), surface-active agent (OP-10) and their content in phosphating bath on PCCs were studied, and the mechanism of action of strong oxidant was analyzed by electrochemical measurement and scanning electron microscopy (SEM).

## 2. Materials and Methods

### 2.1. Sample Preparation

An AZ91D magnesium alloy was used as the substrate material; the content is shown in Table 1. After cutting into a dimension of $30 \times 30 \times 5$ mm and mechanical polishing with up to 2000 grit SiC paper, plate samples were treated in different pretreatments as shown in Table 2 [25–27]. A phosphating bath containing 35 g/L $Ca(NO_3)_2 \cdot 4H_2O$, 20 g/L $NaH_2PO_4 \cdot 2H_2O$, 1~5 g/L $KMnO_4$, 0.5~2.5 g/L $Na_2MoO_4 \cdot 2H_2O$, 0.5~2.5 g/L OP-10 was used to treat the samples after pretreatments, and the process conditions were as follows: phosphating temperature 55 °C, reaction time 10 min, pH 3.0 (regulated with phosphoric acid). After being taken out of the bath, the samples were thoroughly washed using running deionized water, and then dried with cold air for subsequent tests.

**Table 1.** Elemental composition and content of AZ91D magnesium alloy (wt.%).

| Element | Al | Zn | Mn | Si | Cu | Ni | Fe | Mg |
|---|---|---|---|---|---|---|---|---|
| Content | 9.1 | 0.84 | 0.23 | 0.01 | 0.02 | 0.0021 | 0.005 | Margin |

**Table 2.** Formula and operating condition of pretreatment [25–27].

| Sample Number | Formula and Operating Condition | |
|---|---|---|
| No. 1 | | 3 wt.% $H_2SO_4$<br>25 °C, 10 s |
| No. 2 | | 3 wt.% $HNO_3$<br>25 °C, 10 s |
| No. 3 | $Na_3PO_4 \cdot 12H_2O$ 10 g/L<br>NaOH 50 g/L<br>70 °C, 10 min | 5 wt.% HCl<br>25 °C, 10 s |
| No. 4 | | 30 wt.% $H_3PO_4$<br>25 °C, 30 s |
| No. 5 | | $H_3PO_4$ 20 g/L<br>$Na_3PO_4 \cdot 12H_2O$ 12 g/L<br>25 °C, 30 s |
| No. 6 | Sonication cleanout in acetone for 5 min | |

## 2.2. Surface Characterization

The appearances and surface morphologies of the phosphate coatings were observed using a digital camera (Sony ZV-1F, Shenzhen, China) and a scanning electron microscope (SEM, TESCAN MIRA3, Brno, Czech Republic), and the chemical compositions were identified using an energy dispersive spectroscopy (EDS, Oxford Instruments X-Max, Oxford, UK).

## 2.3. Electrical Contact Resistance (ECR) Measurement

The ECR measurement was analyzed with an ohmmeter, two probes were placed vertically on the sample with a distance of 1 cm, and a load of 120 g was applied. The arithmetic average of the measured results in three different areas of the sample was taken as the final test result.

## 2.4. Corrosion Tests

The time of $CuSO_4$ pitting corrosion test of the coating was carried out dropping a drop of 3 wt.% $CuSO_4$ solution on a 1 cm$^2$ area of the sample and recording the time when the droplet changed from blue to black. The arithmetic average of the test results in three different areas of the sample was taken as the final test result.

The potentiodynamic polarization (PDP) and electrochemical impedance spectroscopy (EIS) curves were measured using an electrochemical workstation (CH Instruments Ins CHI660E) in 3.5 wt.% NaCl solution at room temperature. A three-electrode cell was used, with the sample as the working electrode (with 1.0 cm$^2$ exposed area), a platinum electrode as the counter electrode, and a saturated calomel electrode (SCE, +0.242 V vs. SHE) as the reference electrode. Before the test, the sample was immersed in the test solution for about 30 min to stabilize the open circuit potential (OCP). The scanning rate of PDP tests was 5 mV/s, and the scanning range was OCP $\pm$ 0.5 V. The EIS tests were conducted at OCP values, and the scanning frequency range was from 10 mHz to 100 kHz. The data for the PDP and EIS curves were analyzed using tafel extrapolation and ZSimpWin software (ZSimpWin 3.60, http://www.echemsw.com/), respectively.

The immersion test was performed in 3.5 wt.% NaCl solution at room temperature. The appearances of samples were imaged every 12 h.

Each test was repeated at least three times to ensure reproducibility.

## 3. Results and Discussion

### 3.1. Effect of Pretreatment

Due to the strong chemical activity of magnesium alloys [28], the pretreatment before phosphating has a great influence on the phosphate coatings prepared later. Figure 1 shows the appearance of the samples after pretreatment and phosphating, the results of $CuSO_4$ pitting corrosion test and ECR measurement are shown in Figure 2. It is seen that the PCC with good corrosion resistance tend to have poor electrical conductivity. The coatings on sample No. 1, 2 and 3 are incomplete and heterogenous, with low ECR and poor corrosion resistance. The coatings on sample No. 4, 5 and 6 are compact and integral and have good corrosion resistance, but the electrical conductivity of No. 4 and 5 is poor. Therefore, process No. 6, which has both excellent corrosion resistance and good electrical conductivity, is selected as the pretreatment for the phosphate conversion coatings preparation.

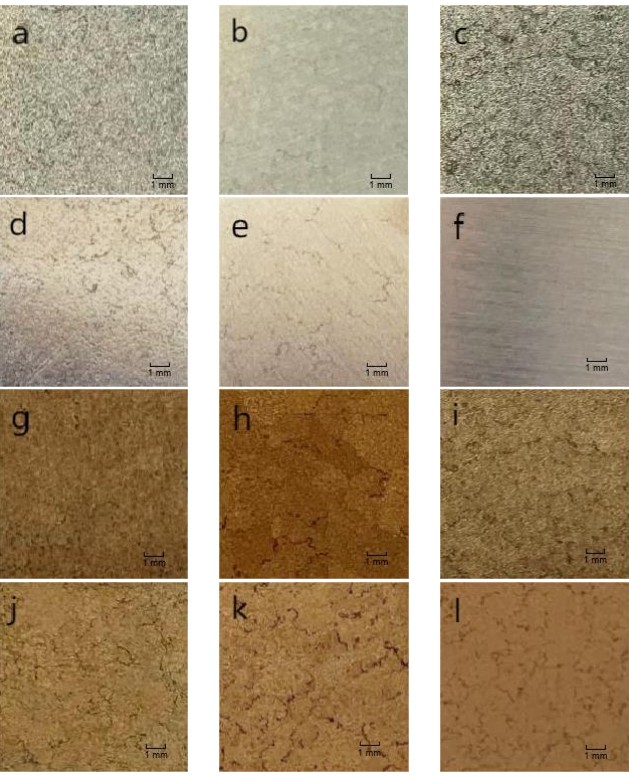

**Figure 1.** Appearance of samples treated with pretreatment and phosphating (**a**–**e**: samples after No. 1–No. 5 pretreatment; **f**: sample No. 6 only ultrasonic cleaned by acetone; **g**–**l**: sample No. 1–No. 6 after phosphating).

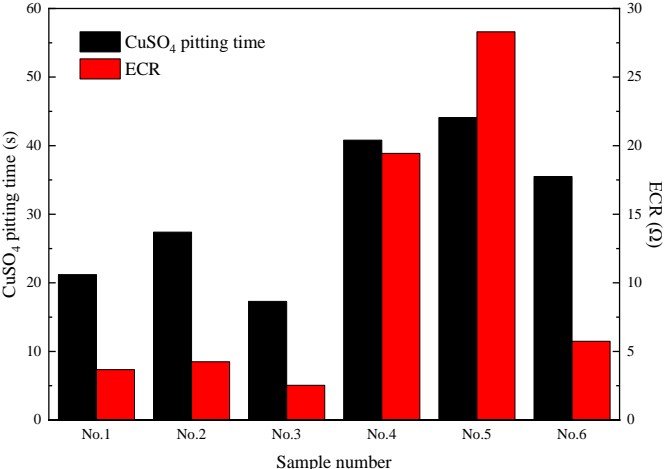

**Figure 2.** Curves of CuSO$_4$ pitting time and ECR of PCCs prepared by different pretreatments.

### 3.2. Effect of Strong Oxidant

In the present study, KMnO$_4$ is applied as the component of strong oxidant in the studied phosphating bath. Figure 3 presents the surface morphologies of samples treated in phosphating bath with KMnO$_4$, and the chemical compositions are shown in Table 3. It is seen from Figure 3b that the coating has a "riverbed" appearance, and numerous irregular bright spots can be seen on the surface. These bright spots are obviously raised, and clear grinding scratches can be seen on the surface, indicating that the coatings on them are very thin, or even that no coatings exist, as shown in Figure 3b. Combined with Table 3, the percentage of Al atoms in bright spots is much higher than that in the surrounding areas, and no Ca and P are detected, indicating that these areas are β-Mg$_{17}$Al$_{12}$ phase, it

is speculated that there is an extremely thin coating composed of Mg, Al and O on the surface, and the surrounding region is α-Mg phase, the main components of the upper coating are phosphates of Ca and Mg and oxides of Mg, Al and Mn. According to previous investigations [29], the action mechanism of $KMnO_4$ is speculated as follows:

$$MnO_4^- + 4H^+ + 3e^- \rightarrow MnO_2 + 2H_2O \tag{1}$$

$$Mg + H_2O - 2e^- \rightarrow MgO + 2H^+ \tag{2}$$

$$2Al + 3H_2O - 6e^- \rightarrow Al_2O_3 + 6H^+ \tag{3}$$

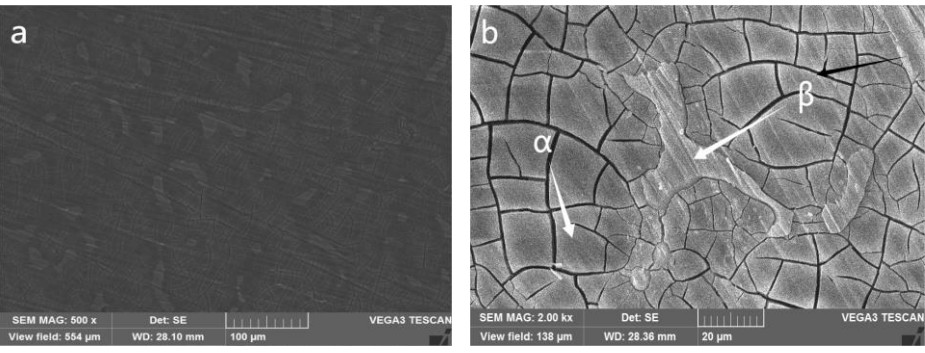

**Figure 3.** Surface morphologies of PCC with $KMnO_4$ (**a**: ×500; **b**: ×2000).

**Table 3.** Chemical composition of PCC with $KMnO_4$ by EDS equipped in SEM.

| Atom (%) | Mg | Al | Ca | Mn | Mo | P | O |
|---|---|---|---|---|---|---|---|
| α phase | 11.6 | 4.0 | 6.1 | 2.6 | 0.5 | 14.5 | 60.6 |
| β phase | 51.38 | 39.24 | — | 0.35 | — | — | 9.03 |

Figure 4 presents the surface morphologies of the coating formed at the phosphating bath free of $KMnO_4$, and the chemical compositions are shown in Table 4. It can be seen from Figure 4a,b that mass flake crystallization generated, and a few protruding β-$Mg_{17}Al_{12}$ phases are also visible, but these β phases are blocked by crystals. Combined with the chemical compositions shown in Table 4 and previous investigations [29], these crystals are presumed to be $CaHPO_4 \cdot 2H_2O$ formed by the deposition of $Ca^{2+}$ due to the rapid rise in pH of solution surrounding the micro cathode β phase caused by the micro-galvanic effect. The following reactions of deposition of $Ca^{2+}$ may occur in the studied phosphating bath:

$$H_2PO_4^{2-} \rightarrow HPO_4^{2-} + H^+ \tag{4}$$

$$Ca^{2+} + HPO_4^{2-} \rightarrow CaHPO_4 \downarrow \tag{5}$$

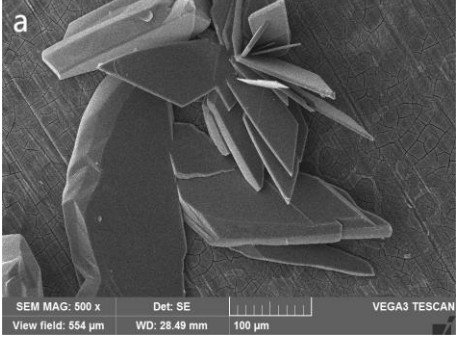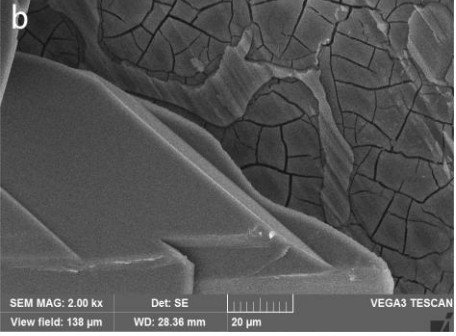

**Figure 4.** Surface morphologies of PCC without $KMnO_4$ (**a**: ×500; **b**: ×2000).

**Table 4.** Chemical composition of coating and crystallization of PCC without KMnO$_4$ by EDS equipped in SEM.

| Atom (%) | Mg | Al | Ca | Mo | P | O |
|---|---|---|---|---|---|---|
| PCC | 2.41 | 0.38 | 12.66 | 0.1 | 16.72 | 67.81 |
| Crystal | 0.42 | — | 21.77 | — | 12.86 | 64.95 |

Table 5 shows the ECR of two coatings measured by the two-point method. The ECR decreases from unmeasurable to 5.74 Ω with the addition of KMnO$_4$. Observe the phenomenon of parkerising, the number of bubbles generated by the reaction in the phosphating bath containing KMnO$_4$ is significantly reduced compared with that without KMnO$_4$. Combined with the surface morphologies above, it is seen that the introduction of KMnO$_4$ can effectively slow down the hydrogen evolution reaction in the reaction, inhibit the formation of CaHPO$_4$·2H$_2$O crystals and form an extremely thin oxide coating on the β phase, which finally protrude from the surface to become "conductive spots", and the conductivity of the phosphate conversion coating is improved.

**Table 5.** The ECR of phosphate conversion coating with and without KMnO$_4$.

| Composition of Phosphating Bath | ECR (Ω) |
|---|---|
| Without KMnO$_4$ | — |
| With KMnO$_4$ | 5.74 |

The results of the electrochemical tests for the PCCs formed at phosphating bath with different content of KMnO$_4$ are shown in Figure 5. The main fitting results of the PDP and EIS curves are summarized in Tables 6 and 7, respectively, and the equivalent electrical circuit (EEC) model used to fit the EIS curves is shown in Figure 6. R$_s$ is the resistance of solution, R$_1$, CPE$_1$ and R$_2$, CPE$_2$ represent the resistance and capacitance of coating and double electric layer, respectively.

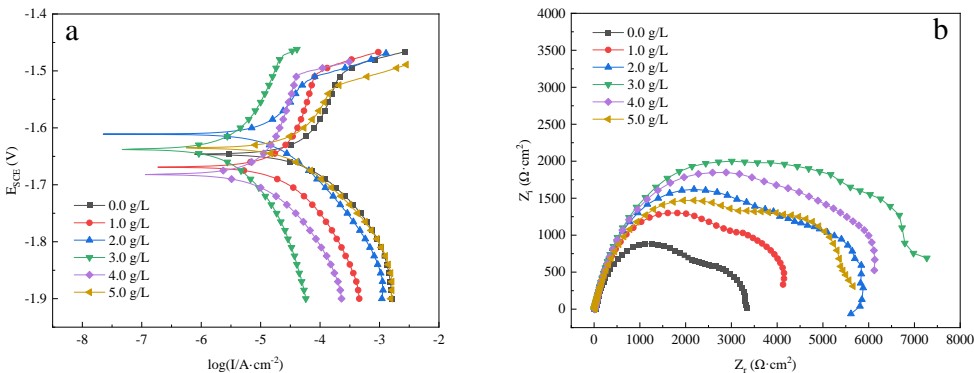

**Figure 5.** PDP and EIS curves of the PCCs obtained at different contents of KMnO$_4$ (**a**: PDP curves; **b**: EIS curves).

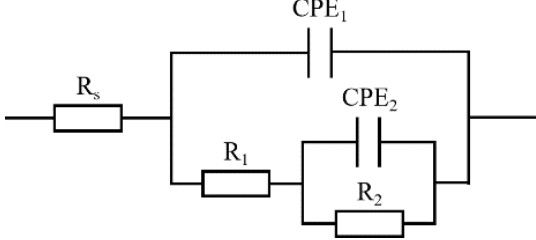

**Figure 6.** Equivalent circuit for EIS curve.

**Table 6.** Electrochemical parameters derived from the PDP curves of the PCCs obtained at different contents of KMnO$_4$.

| Content of KMnO$_4$ (g/L) | E$_{corr}$ (V) | i$_{corr}$ (A/cm$^2$) |
|:---:|:---:|:---:|
| 0.0 | −1.646 | $5.27 \times 10^{-5}$ |
| 1.0 | −1.669 | $3.61 \times 10^{-5}$ |
| 2.0 | −1.611 | $2.04 \times 10^{-5}$ |
| 3.0 | −1.638 | $9.62 \times 10^{-6}$ |
| 4.0 | −1.682 | $1.39 \times 10^{-5}$ |
| 5.0 | −1.635 | $2.86 \times 10^{-5}$ |

**Table 7.** Fitting results of EIS curves of the PCCs obtained at different contents of KMnO$_4$.

| Content of KMnO$_4$ (g/L) | 0.0 | 1.0 | 2.0 | 3.0 | 4.0 | 5.0 |
|:---:|:---:|:---:|:---:|:---:|:---:|:---:|
| R$_s$ ($\Omega \cdot$cm$^2$) | 30.95 | 8.779 | 15.96 | 8.011 | 8.965 | 10.64 |
| CPE$_1$/($\mu$S·s$^{-n}$·cm$^{-2}$) | $2.697 \times 10^{-5}$ | $1.484 \times 10^{-5}$ | $1.215 \times 10^{-5}$ | $1.362 \times 10^{-5}$ | $1.375 \times 10^{-5}$ | $1.458 \times 10^{-5}$ |
| n$_1$ | 0.8091 | 0.7914 | 0.8403 | 0.7817 | 0.7867 | 0.7759 |
| R$_1$ ($\Omega \cdot$cm$^2$) | 2486 | 3744 | 4478 | 5956 | 5798 | 4335 |
| CPE$_2$ ($\mu$S·s$^{-n}$·cm$^{-2}$) | $9.589 \times 10^{-4}$ | $5.534 \times 10^{-4}$ | $3.411 \times 10^{-4}$ | $2.598 \times 10^{-4}$ | $6.991 \times 10^{-4}$ | $3.071 \times 10^{-4}$ |
| n$_2$ | 0.9244 | 0.8021 | 0.8947 | 0.8736 | 0.7641 | 0.8768 |
| R$_2$ ($\Omega \cdot$cm$^2$) | 780 | 675 | 1404 | 1357 | 568 | 1378 |
| R$_1$ + R$_2$ ($\Omega \cdot$cm$^2$) | 3266 | 4419 | 5882 | 7313 | 6357 | 5713 |

The results show that adding KMnO$_4$ into the phosphating bath can improve the corrosion resistance of the PCCs. It can be seen from Figure 5a and Table 6 that with the content of KMnO$_4$ increasing, the corrosion current (i$_{corr}$) of the coating decreases first and then increases, and the PCC exhibit the minimum i$_{corr}$ of $9.62 \times 10^{-6}$ A/cm$^2$ when the content of KMnO$_4$ is 3.0 g/L. As for the EIS curves in Figure 5b, the Nyquist curves all have similar characteristics and consist of two capacitive loops. The high-frequency capacitive loop is related to the process of charge transfer from magnesium alloy matrix to solution double layer during corrosion, while the low-frequency capacitive loop represents the process of Mg$^{2+}$ diffusion to the sample surface. The order of dimension of the high-frequency capacitive loop is as follows: 3.0 g/L > 4.0 g/L > 2.0 g/L > 5.0 g/L > 1.0 g/L > 0.0 g/L. Combined with the data in Table 7, the coating resistance R$_1$ reaches the maximum of 5956 $\Omega \cdot$cm$^2$ at the content of KMnO$_4$ is 3.0 g/L. Therefore, the optimal content of KMnO$_4$ in the studied phosphating bath to form PCCs with the best corrosion resistance is 3.0 g/L.

In addition to corrosion resistance, low ECR is also the focus of conductive and corrosion-resistant phosphate conversion coatings. Therefore, CuSO$_4$ pitting corrosion test and contact resistance are used as the evaluation basis for corrosion resistance and electrical conductivity, respectively, to comprehensively screen the appropriate content of KMnO$_4$.

Figure 7 presents the results of the CuSO$_4$ pitting corrosion test and ECR measurement of PCCs with different KMnO$_4$ contents. It is seen that the concentration of KMnO$_4$ has a significant influence on the CuSO$_4$ pitting time and ECR. With the increase in KMnO$_4$ concentration, the pitting time of the coatings first increases and then decreases, which is consistent with the electrochemical test results above, and the ECR shows a trend of decrease, with the magnitude of the decreasing has lessened. In the formation of PCC, MnO$_4^-$ will react with H$^+$, so that the pH between the solution and the alloy increases, thus promoting the deposition of phosphate and accelerating the formation of coating. However, when the content of KMnO$_4$ is too high, the surface of Mg alloy will be passivated, which will inhibit the subsequent deposition reaction, and the thickness of the coating will become thinner, the corrosion resistance will decrease, thus improving the electrical

conductivity [30–32]. When the concentration of KMnO$_4$ is 3.0 g/L, the coating has the longest pitting time and a low ECR. With the further increase in the concentration, the pitting time is significantly shortened, but the decrease in the ECR is not obvious. Therefore, the optimum content of KMnO$_4$ in the studied phosphating bath is 3.0 g/L.

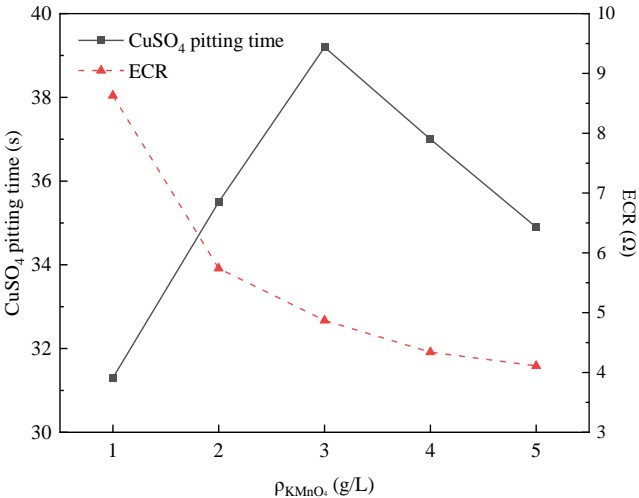

**Figure 7.** Curves of CuSO$_4$ pitting time and ECR of PCCs formed at different KMnO$_4$ content.

### 3.3. Effect of Additive

In the present study, Na$_2$MoO$_4$ is applied as the component of additive in the studied phosphating bath. Figure 8 presents the appearances of samples treated in phosphating bath with and without Na$_2$MoO$_4$ at various times during the 48 h immersion test, and the time interval of each photograph is 12 h. It is seen that both samples exhibited obvious pitting corrosion. The sample without Na$_2$MoO$_4$ showed obvious corrosion points at 24 h, while the sample containing Na$_2$MoO$_4$ showed corrosion points after 36 h. After the 48 h immersion test, the corroded area of both samples was intensified. However, the corroded area on the surface of sample with Na$_2$MoO$_4$ was less than that of sample without Na$_2$MoO$_4$. The result shows that adding Na$_2$MoO$_4$ into the phosphating bath can effectively improve the corrosion resistance of the coatings.

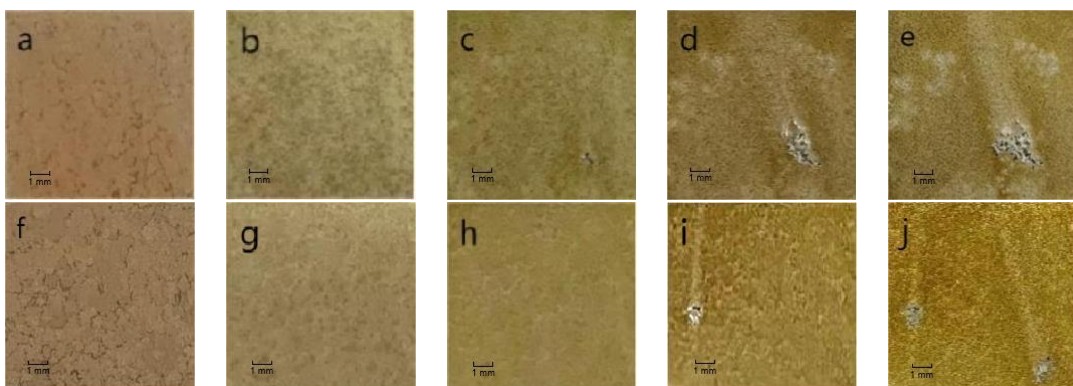

**Figure 8.** Appearances of samples treated in phosphating bath with and without Na$_2$MoO$_4$ at various times during the 48 h immersion test (**a**–**e**: without Na$_2$MoO$_4$, 0–48 h; **f**–**j**: with Na$_2$MoO$_4$, 0–48 h).

Figure 9 shows the results of electrochemical tests for the PCCs formed in the phosphating bath with different contents of Na$_2$MoO$_4$. The main fitting results of the PDP and EIS curves are summarized in Tables 8 and 9, respectively. MoO$_4^{2-}$ can form insoluble CaMoO$_4$ with Ca$^{2+}$. A small amount of Na$_2$MoO$_4$ can increase the nucleation site in the phosphating process to promote the deposition of Ca$^{2+}$ and accelerate the coating formation. However, when the concentration of Na$_2$MoO$_4$ is too high, cracks in the coating will be

thickened and many uneven coarse crystals will be generated on the surface, leading to the decline in corrosion resistance [33]. It can be seen from Figure 9 that with the increasing concentration of $Na_2MoO_4$, the $i_{corr}$ of the coating decreases first and then increases, while the dimension of the high-frequency capacitive loop increases first and then decreases. According to the data in Tables 8 and 9, when the content of $Na_2MoO_4$ is 1.5 g/L, the $i_{corr}$ of the PCC is the lowest ($3.74 \times 10^{-6}$ A/cm$^2$), and R1 reaches the maximum (7841 $\Omega \cdot$cm$^2$). Therefore, the optimal content of $Na_2MoO_4$ in the studied phosphating bath to form PCCs with the best corrosion resistance is 1.5 g/L.

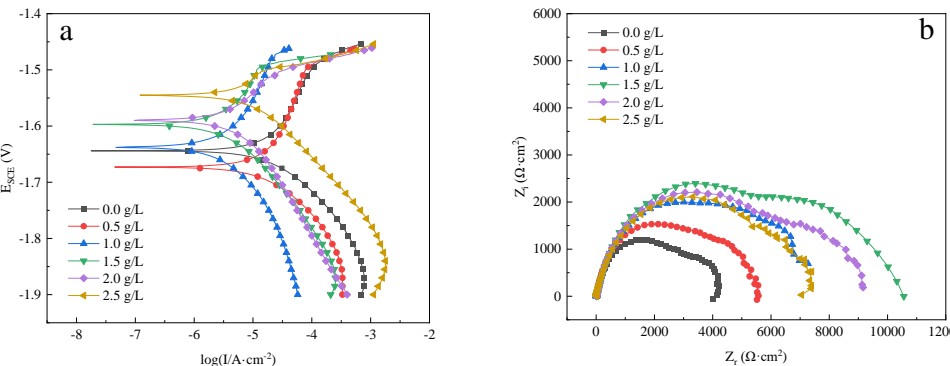

**Figure 9.** PDP and EIS curves of the PCCs obtained at different contents of $Na_2MoO_4$ (**a**: PDP curves; **b**: EIS curves).

**Table 8.** Electrochemical parameters derived from the PDP curves of the PCCs obtained at different contents of $Na_2MoO_4$.

| Content of Na$_2$MoO$_4$ (g/L) | E$_{corr}$ (V) | i$_{corr}$ (A/cm$^2$) |
|:---:|:---:|:---:|
| 0.0 | −1.644 | $2.02 \times 10^{-5}$ |
| 0.5 | −1.673 | $1.87 \times 10^{-5}$ |
| 1.0 | −1.638 | $9.62 \times 10^{-6}$ |
| 1.5 | −1.597 | $3.74 \times 10^{-6}$ |
| 2.0 | −1.590 | $7.53 \times 10^{-6}$ |
| 2.5 | −1.545 | $1.02 \times 10^{-5}$ |

**Table 9.** Fitting results of EIS curves of the PCCs obtained at different contents of $Na_2MoO_4$.

| Content of Na$_2$MoO$_4$ (g/L) | 0.0 | 0.5 | 1.0 | 1.5 | 2.0 | 2.5 |
|:---|:---:|:---:|:---:|:---:|:---:|:---:|
| R$_s$ ($\Omega \cdot$cm$^2$) | 11.34 | 10.72 | 8.011 | 9.342 | 12.73 | 12.1 |
| CPE$_1$ ($\mu$S$\cdot$s$^{-n}\cdot$cm$^{-2}$) | $1.665 \times 10^{-5}$ | $1.481 \times 10^{-5}$ | $1.362 \times 10^{-5}$ | $1.575 \times 10^{-5}$ | $1.79 \times 10^{-5}$ | $1.859 \times 10^{-5}$ |
| n$_1$ | 0.8395 | 0.827 | 0.7817 | 0.7392 | 0.8423 | 0.8357 |
| R$_1$ ($\Omega \cdot$cm$^2$) | 3226 | 4289 | 5956 | 7841 | 6638 | 6165 |
| CPE$_2$ ($\mu$S$\cdot$s$^{-n}\cdot$cm$^{-2}$) | $4.725 \times 10^{-4}$ | $3.455 \times 10^{-4}$ | $2.598 \times 10^{-4}$ | $3.636 \times 10^{-4}$ | $6.126 \times 10^{-4}$ | $8.546 \times 10^{-4}$ |
| n$_2$ | 0.9179 | 0.8701 | 0.8736 | 0.8127 | 0.8335 | 0.8314 |
| R$_2$ ($\Omega \cdot$cm$^2$) | 979 | 1187 | 1357 | 2278 | 2166 | 962 |
| R$_1$ + R$_2$ ($\Omega \cdot$cm$^2$) | 4205 | 5476 | 7313 | 10,119 | 8804 | 7127 |

Figure 10 shows the results of the CuSO$_4$ pitting corrosion test and ECR measurement of PCCs with different $Na_2MoO_4$ contents. It is seen that the concentration of $Na_2MoO_4$ has a great influence on the CuSO$_4$ pitting time, which increases first and then decreases with the increase in the content, reaching the maximum at 1.5 g/L, which is consistent with the results of electrochemical tests. However, the concentration of $Na_2MoO_4$ has

little effect on the ECR, which shows an upward trend, and the difference between the minimum 0.0 g/L and the maximum 2.5 g/L is about 1.5 Ω. Thus, 1.5 g/L is the optimum content of $Na_2MoO_4$ in the studied phosphating bath considering corrosion resistance and electrical conductivity.

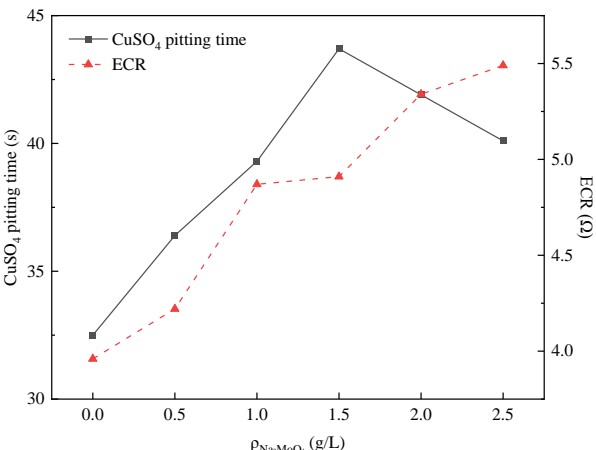

**Figure 10.** Curves of $CuSO_4$ pitting time and ECR of PCCs formed at different $Na_2MoO_4$ content.

### 3.4. Effect of Surface-Active Agent

In the present study, OP-10 is applied as the component of surface-active agent in the studied phosphating bath. Figure 11 presents the surface morphologies of PCCs formed in the phosphating bath with and without OP-10. It is seen that no matter whether OP-10 is contained in the phosphating bath or not, both coatings have a "dry riverbed" appearance. However, the cracks on the PCC containing OP-10 are significantly fewer than that without OP-10, and the crack width is also decreased. According to previous investigations [34], the composition of OP-10 is alkylphenol ethoxylates (APEO), which can reduce the tension between the metal and the solution interface, allowing the phosphating bath to moisten the surface rapidly, and can also reduce the adhesion ability of bubbles generated by the reaction, making them more easily separable from the sample surface, so that the compactness of the coating is improved.

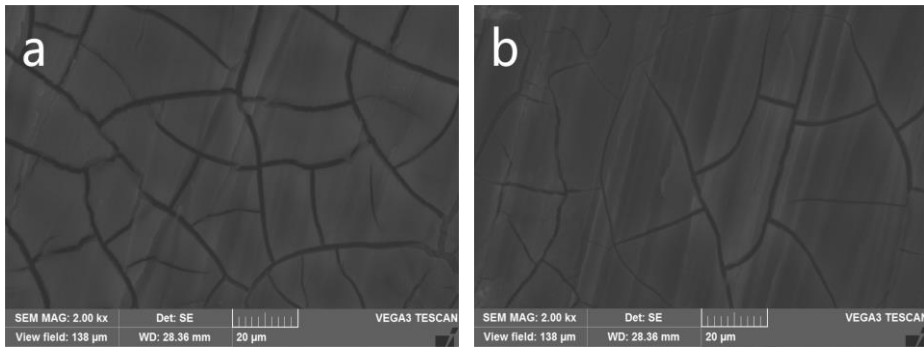

**Figure 11.** Surface morphologies of PCCs with and without OP-10 (**a**: without OP-10 ×2000; **b**: with OP-10 ×2000).

Figure 12 shows the results of electrochemical tests for the PCCs formed at phosphating bath with different contents of OP-10. The main fitting results of the PDP and EIS curves are summarized in Tables 10 and 11, respectively. It is seen that the $i_{corr}$ of the coating decreases in the following order: 0.0 g/L > 2.5 g/L > 2.0 g/L >1.5 g/L > 0.5 g/L > 1.0 g/L. The $i_{corr}$ reaches the minimum ($3.74 \times 10^{-6}$ A/cm$^2$) when OP-10 concentration is 1.0 g/L, and the coating resistance $R_1$ of the PCC reaches the maximum (7841 Ω·cm$^2$). Therefore,

the optimal content of OP-10 in the studied phosphating bath to form PCCs with the best corrosion resistance is 1.0 g/L.

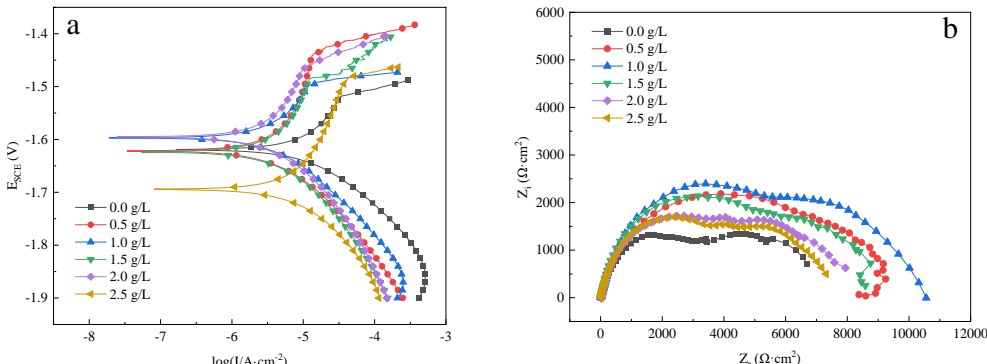

**Figure 12.** PDP and EIS curves of the PCCs obtained at different contents of OP-10 (**a**: PDP curves; **b**: EIS curves).

**Table 10.** Electrochemical parameters derived from the PDP curves of the PCCs obtained at different contents of OP-10.

| Content of OP-10 (g/L) | $E_{corr}$ (V) | $i_{corr}$ (A/cm$^2$) |
|---|---|---|
| 0.0 | −1.619 | $1.35 \times 10^{-5}$ |
| 0.5 | −1.622 | $7.52 \times 10^{-6}$ |
| 1.0 | −1.597 | $3.74 \times 10^{-6}$ |
| 1.5 | −1.624 | $8.19 \times 10^{-6}$ |
| 2.0 | −1.595 | $9.45 \times 10^{-6}$ |
| 2.5 | −1.694 | $1.01 \times 10^{-5}$ |

**Table 11.** Fitting results of EIS curves of the PCCs obtained at different contents of OP-10.

| Content of OP-10 (g/L) | 0.0 | 0.5 | 1.0 | 1.5 | 2.0 | 2.5 |
|---|---|---|---|---|---|---|
| $R_s$ ($\Omega \cdot cm^2$) | 10.72 | 14.48 | 9.342 | 11.26 | 10.99 | 12.07 |
| $CPE_1$ ($\mu S \cdot s^{-n} \cdot cm^{-2}$) | $1.484 \times 10^{-5}$ | $1.748 \times 10^{-5}$ | $1.575 \times 10^{-5}$ | $1.867 \times 10^{-5}$ | $1.611 \times 10^{-5}$ | $1.305 \times 10^{-5}$ |
| $n_1$ | 0.7677 | 0.8481 | 0.7392 | 0.838 | 0.7655 | 0.7925 |
| $R_1$ ($\Omega \cdot cm^2$) | 3944 | 7468 | 7841 | 6126 | 5268 | 4957 |
| $CPE_2$ ($\mu S \cdot s^{-n} \cdot cm^{-2}$) | $3.181 \times 10^{-4}$ | $6.223 \times 10^{-4}$ | $3.636 \times 10^{-4}$ | $4.685 \times 10^{-4}$ | $3.538 \times 10^{-4}$ | $2.905 \times 10^{-4}$ |
| $n_2$ | 0.8529 | 0.7445 | 0.8127 | 0.8662 | 0.8763 | 0.8331 |
| $R_2$ ($\Omega \cdot cm^2$) | 2804 | 1347 | 2278 | 2292 | 2522 | 2261 |
| $R_1 + R_2$ ($\Omega \cdot cm^2$) | 6748 | 8815 | 10,119 | 8418 | 7790 | 7218 |

Figure 13 shows the results of the CuSO$_4$ pitting corrosion test and ECR measurement of PCCs with different OP-10 contents. It is seen that the ECR of coatings fluctuated between 4.4 Ω and 5.2 Ω, indicating that the concentration of OP-10 had no significant effect on electrical conductivity, while the pitting time increases first and then decreases with the increase in concentration, reaching the maximum at 1.0 g /L, which is consistent with the electrochemical test results. Therefore, the optimum content of OP-10 in the studied phosphating bath is 1.0 g/L.

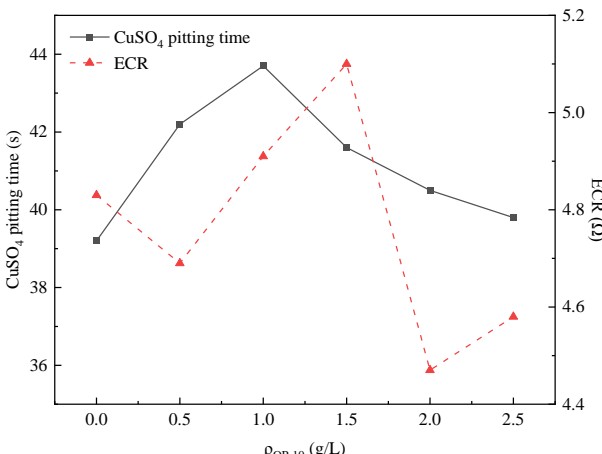

**Figure 13.** Curves of CuSO₄ pitting time and ECR of PCCs formed at different OP-10 content.

*3.5. Performance Tests Results of PCC*

From the results above, the optimum content for $KMnO_4$, $Na_2MoO_4$ and OP-10 in phosphating bath is 3.0 g/L, 1.5 g/L and 1.0 g/L, respectively.

Figure 14 shows the results of the electrochemical tests for the uncoated AZ91D magnesium alloy and the PCC formed at the optimum phosphating bath formulation. The main fitting results of the PDP are summarized in Table 12. The results show that PCC can provide good protection for the magnesium alloy. It can be seen from Figure 14a and Table 12 that the coated alloy has more positive $E_{corr}$ and lower $i_{corr}$ than the uncoated one. The $E_{corr}$ positively shifts about 27 mV, and the $i_{corr}$ decreases by more than 1 order of magnitude.

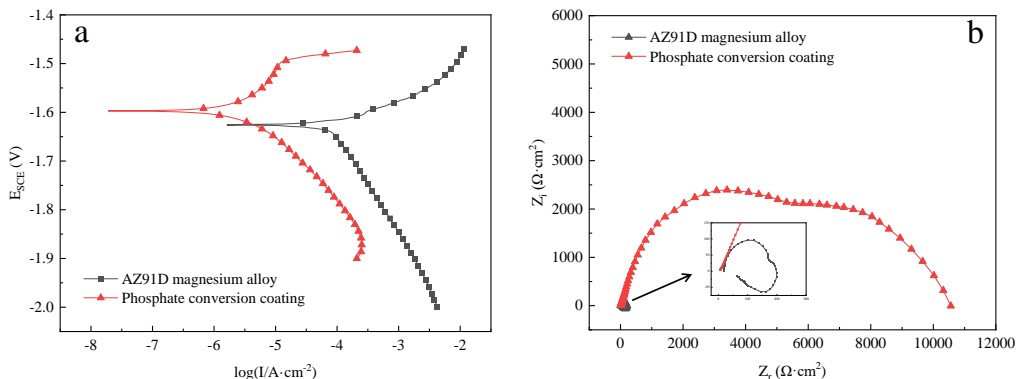

**Figure 14.** PDP and EIS curves of uncoated AZ91D magnesium alloy and PCC formed at the phosphating bath at the optimum condition (**a**: PDP curves; **b**: EIS curves).

**Table 12.** Electrochemical parameters derived from the PDP curves of uncoated AZ91D magnesium alloy and PCC formed at the phosphating bath at the optimum condition.

| Sample | $E_{corr}$ (V) | $i_{corr}$ (A/cm²) |
|---|---|---|
| Phosphate conversion coating | −1.597 | $3.74 \times 10^{-6}$ |
| AZ91D magnesium alloy | −1.626 | $7.84 \times 10^{-5}$ |

As for the EIS curves in Figure 14b, the Nyquist curve of the PCC consisted of two capacitive loops, while that of the uncoated alloy has an obvious inductive loop in the low-frequency region. The shape of the impedance spectrum describes the type of electrochemical reactions that occur at the electrode surface. The low-frequency inductive loop corresponds to the adsorption process of $Mg^{2+}$ on the alloy surface. A fully covered PCC

can block the contact between alloy and the test solution, thus hindering the generation and adsorption of $Mg^{2+}$ on the surface. Therefore, no inductive loop appears in the Nyquist curve of the PCC.

Figure 15 presents the surface morphologies and the chemical compositions of PCC formed at the phosphating bath at the optimum condition. It is seen that the coating has a "dry riverbed" appearance and is covered completely. Clear irregular bright spots can be seen, which are raised β phases according to the previous results. The β phases combined with the ECR of the PCC is 4.91 Ω, demonstrating that β phases act as a conductive spot to achieve electrical conductivity.

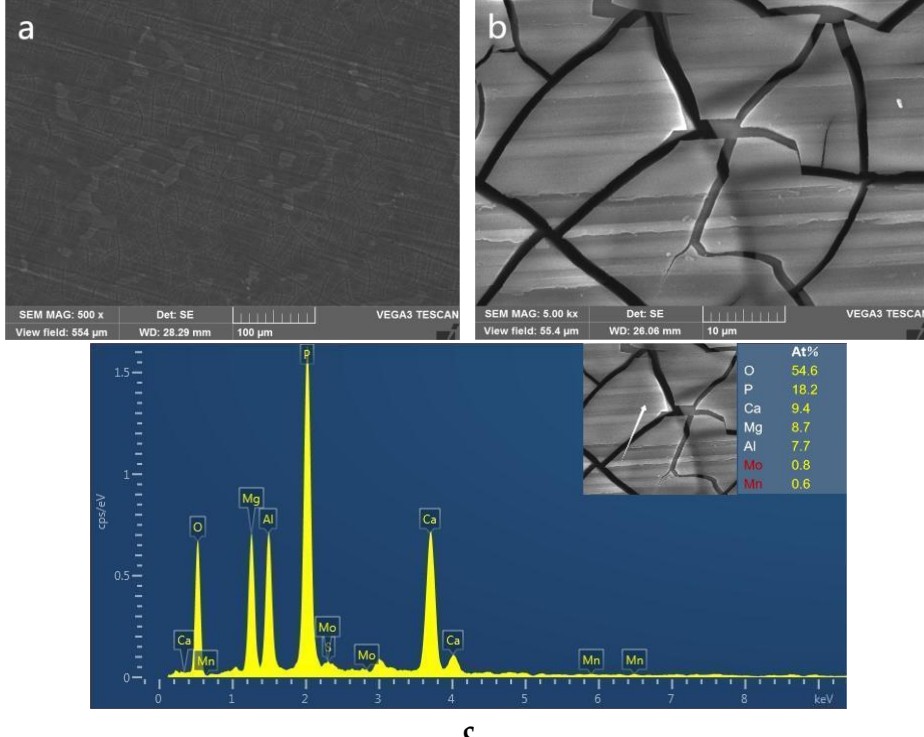

**Figure 15.** Surface morphologies and the chemical compositions of PCC formed at the phosphating bath at the optimum condition (**a**: ×500; **b**: ×2000; **c**: EDS).

Obvious scratches are visible on the surface, showing that the coating is thin, so that it cannot effectively cover the polishing marks. However, the bilayer structure can be clearly seen from the coating upon α phases in Figure 15b, which is consistent with the structure of Ca phosphate conversion coatings in previous investigations [35], indicating that the PCC has good corrosion resistance.

According to the chemical compositions shown in Figure 15c, the PCC consists of Ca, Mg, Al, O, P and a small amount of Mn and Mo, and it is speculated that the main coating-forming substances are phosphate of Ca and Mg and oxides of Mg and Al.

## 4. Conclusions

The conductive and corrosion-resistant phosphate conversion coating was prepared upon the surface of AZ91D magnesium alloy. The PCC was completely covered, and the main coating-forming substances were phosphate of Ca and Mg and oxides of Mg and Al. The coating on α phases had bilayer structure and the β phases protruded as conductive spots, which provided high corrosion resistance and low ECR.

The presence of strong oxidant $KMnO_4$ formed an extremely thin passive film on the protruded β phases and inhibited the formation of phosphate crystals to prevent the conductive spots from being obscured.

The optimum content for KMnO$_4$, Na$_2$MoO$_4$ and OP-10 in phosphating bath was 3.0 g/L, 1.5 g/L and 1.0 g/L, and the pretreatment before phosphating was sonication cleanout in acetone for 5 min after mechanical polishing.

**Author Contributions:** Conceptualization, Y.X. and S.Z.; methodology, Y.X.; software, Y.X.; validation, S.Z., J.D. and Y.X.; formal analysis, S.Z. and T.Z.; investigation, Y.X.; resources, S.Z.; data curation, Y.X.; writing—original draft preparation, Y.X.; writing—review and editing, Y.X.; visualization, L.L. and S.Z.; supervision, Q.L.; project administration, Y.X. and S.Z.; funding acquisition, S.Z. All authors have read and agreed to the published version of the manuscript.

**Funding:** This work was supported by the Key project of National Natural Science Foundation of China: Research basis of "corrosion-functional" integrated protective coating on magnesium alloys (U21A2045). the National Natural Science Foundation of China (NSFC), "Study of conductive-corrosion resistant chemical conversion coating of magnesium alloy" (51771050).

**Data Availability Statement:** Not applicable.

**Conflicts of Interest:** The authors declare no conflict of interest.

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
