# Peer review of "Preparation of Conductive and Corrosion Resistant Phosphate Conversion Coating on AZ91D Magnesium Alloy"

_coatings, doi:10.3390/coatings13101706_

Round 1

Reviewer 1 Report

The presence of KMnO4 inhibited the 17 formation of phosphate crystals and made the β phases protrude from the surface to form conductive spots, which improved the conductivity of PCCs. Some comments given below.

1.      Nothing truly unique in its current state. Because of the lack of a novel, the current submission looks to be a replication or modified work. The authors must describe their novel in detail. This work should be rejected owing to a major issue.

2.      Line 24-25, related to several application of magnesium alloy, please giving additional relevant reference as follows: https://doi.org/10.3390/su15010823

3.      Line 68, please recheck the composition of AZ91D used. And where it is obtained?

4.      Line 78, why the concentration of H2DO4 should be 3, 5, and 30%? Please give the rationalisation.

5.      Line 95-96, the detail procedure for electrochemical impedance spectroscopy with the chemical reaction should be given.

6.      Line 110, where is the scale> I think the authors forgot to put it.

7.      Line 128, giving comprehensive discussion of the trend for CuSO4 pitting time and ECR of PCCs.

-

Author Response

First of all, thank you very much for taking time out of your busy schedule to review my paper. Next I will answer your comments one by one.

  1. Line 24-25, related to several application of magnesium alloy, please giving additionalrelevant reference as follows: https://doi.org/10.3390/su15010823.

         A: Additions have been made.

  1. Line 68, please recheck the composition of AZ91D used. And where it is obtained?

A: Already confirmed.

  1. Line 78. why the concentration of H2DO4 should be 3. 5. and 30%? Please give the rationalisation.

A: The relevant literature has been supplemented.

  1. Line 95-96. the detail procedure for electrochemical impedance spectroscopy with the chemical reaction should be given.

A: The test procedure of electrochemical impedance spectroscopy has been given in this paper.

  1. Line 110, where is the scale> I think the authors forgot to put it.

A: Additions have been made.

  1. Line 128, giving comprehensive discussion of the trend for CuSO4 pitting time and ECR of PCCs.

A: Additions have been made.

7.      Line 128, giving comprehensive discussion of the trend for CuSO4 pitting time and ECR of PCCs.

A: Additions have been made.

Reviewer 2 Report

The submission is very interesting. One of the best I had the pleasure to review this year. There are some minor issues that you can deal with in no time and be ready for publishing.

The references should not be placed in the body of the text in the upper index like that [1]. Instead, they should look like this [1].

Maybe provide a table with the complete chemical composition of the used AZ91D alloy. In line 68 you have presented only selected elements.

In Table 1 maybe instead of ‘Number’ it would be more clear to write ‘sample number’.

Can you really connect the data points presented in Figure 2? Those are the results of different samples obtained by the utilisation of different agents. I do not think there is any variable changing in a way that could justify such a presentation. Also in the abscissa axis maybe again it would be better to write ‘Sample number’ instead of  ‘number’.

The formatting of the conclusion looks odd. Please change it to look like in the previous parts of the text.

Please change the formatting of the references to be following the coatings template.

There is 1 reference from 2022, also 2 from 2022, the next newest is from 2019. Please add more recently published references. Maybe some non-Chinese ones.

Overall, good job!

Author Response

First of all, thank you very much for taking time out of your busy schedule to review my paper. Next I will answer your comments one by one.

  1. The references should not be placed in the body of the text in the upper index like that [1]nstead, they should look like this [1].

A: Modifications have been made.

  1. Maybe provide a table with the complete chemical composition of the used AZ91D alloy. In line68 you have presented only selected elements

A: Additions have been made.

  1. In Table 1 maybe instead of Number' it would be more clear to write 'sample number'.

A: Modifications have been made.

  1. Can you really connect the data points presented in Figure 2? Those are the results of different samples obtained by the utilisation of different aents. I do not think there is any variable changing in a way that could justify such a presentation. Also in the abscissa axis maybe again it would be better to write Sample number' instead of number”

A: Modifications have been made.

  1. The formatting of the conclusion looks odd. Please change it to look like in the previous parts ofthe text.

A: Modifications have been made.

  1. Please change the formatting of the references to be following the coatings template

There is 1 reference from 2022, also 2 from 2022, the next newest is from 2019. Please addmore recently published references.Maybe some non-Chinese ones.

A: Modifications have been made.

Reviewer 3 Report

11.  Line 133. In the description, the Authors have, according to the reviewer, made a mistake - instead of Fig.3a, it should be 3b.

22.    Line 150. In the description, the Authors have, according to the reviewer, made a mistake - instead of Table 2, it should be Table 3.

33.    Line 151. In the description, the Authors have, according to the reviewer, made a mistake - instead of Fig.4a and 4d, it should be Fig.4a and 4b.

44.    In Table 2, the Authors presented the chemical composition of PCC with KMnO4 by EDS equipped in SEM, whereas in Fig. 3: surface morphologies of PCC with KMnO4 (a: ×500; b: ×2000). The article does not indicate from which places in the photographs presented in Fig. 3 the EDS chemical analysis was conducted. Please mark these places on Fig. 3.

55.    Please remove Table 4 from the article as it does not bring anything new, especially in light of the fact that in Fig. 7 the Authors include curves of CuSO4 pitting time and ECR of PCCs formed at different KMnO4 content.

66.   Please complete the description of Fig.7 with a discussion of the reasons for the decrease in corrosion resistance after exceeding the optimal content of KMnO4.

77.    For Fig. 3 and Fig. 11, please complement the discussion of the results for these figures with SEM surface studies. Moreover, the article lacks a similar figure for the Na2MoO4 supplement and SEM surface studies for this supplement. Please complete it.

88.    In Fig. 8, the Authors indicated: Appearances of samples treated in the phosphating bath with and without Na2MoO4 at various times during the 48-hour immersion test (a-e: without Na2MoO4, 0-48 h; f-j: with Na2MoO4, 0-48 h). Please mark the magnification for which the microstructure photographs were taken and provide the time intervals at which each photograph was taken.

99.    In section 3.5 of the article, the Authors analyze the properties of PCC. Please explain what type of coating it is, with the additive or without. Since according to the presented results, properties such as Ecorr and icor are the same as in the case of a bath with the OP-10 additive.

110. The conclusions lack a conclusion regarding which of the analyzed additives is the best in terms of technical, quality, and economic aspects.

111.   The first sentence included in conclusion no. 2 is a repetition of conclusion no. 1, please revise.

Author Response

First of all, thank you very much for taking time out of your busy schedule to review my paper. Next I will answer your comments one by one.

  1. Line 133. In the description, the Authors have, according to the reviewer, made a mistakeSuggestions for Authorsinstead of Fig.3a, it should be 3b.

Line 150. In the description, the Authors have, according to the reviewer, made a mistakeinstead of Table 2, it should be Table 3.

Line 151. n the description, the Authors have, according to the reviewer, made a mistakeinstead of Fig.4a and 4d, it should be Fig.4a and 4b.

A: Modifications have been made.

  1. In Table 2, the Authors presented the chemical composition of PCC with KMnO4 by EDSequipped in SEM, whereas in Fig. 3: surface morphologies of PCC with KMnO4 (a: x500; bx2000) The article does not indicate from which places in the photographs presented in Fig. 3the EDS chemical analysis was conducted. Please mark these places on Fig. 3.

A: Modifications have been made.

  1. Please remove Table 4 from the article as it does not bring anything new, especially in light ofthe fact that in Fig. 7 the Authors include curves of CuSO4 pitting time and ECR of PCCs formedat different KMnO4 content.

A: Table 4 (now is Table 5) is an important part of introducing the effects of KMnO4, Fig. 7 is used to screen for optimal content. The two functions are different, and I hope that Table 4 can be retained.

  1. Please complete the description of Fiq.7 with a discussion of the reasons for the decrease incorrosion resistance after exceeding the optimal content of KMnO4.

A: Additions have been made.

  1. ln Fig. 8, the Authors indicated: Appearances of samples treated in the phosphating bath withand without Na2Mo04 at various times during the 48-hour immersion test (a-e: withoutNa2Mo04,0-48 h; f: with Na2Mo04, 0-48 h). Please mark the magnification for which themicrostructure photographs were taken and provide the time intervals at which each photographwas taken.

A: Additions have been made.

  1. In section 3.5 of the article, the Authors analyze the properties of PCC. Please explain what typeof coating it is, with the additive or without Since according to the presented results, propertiessuch as Ecorr and icor are the same as in the case of a bath with the OP-10 additive.

A: It is the results of the PCC formed at the optimum phosphating bath formulation. Modifications have been made.

  1. The conclusions lack a conclusion regarding which of the analyzed additives is the best in termsof technical.quality, and economic aspects. The first sentence included in conclusion no. 2 is a repetition of conclusion no.1.please revise.

A: This paper does not compare the influence of the three additives on the PCCS, and the conclusion has been modified.

Round 2

Reviewer 1 Report

Reviewers greatly appreciate the efforts that have been made by the author to improve the quality of their articles after peer review. I reread the author's manuscript and further reviewed the changes made along with the responses from previous reviewers' comments. Unfortunately, the authors failed to make some of the substantial improvements they should have made making this article not of decent quality with biased, not cutting-edge updates on the research topic outlined. In addition, the author also failed to address the previous reviewer's comments, especially on comments number 1 (lack of novel), 2 (not incorporate the literature), and 7 (not a comprehensive discussion). Thank you very much for the opportunity to read the author's current work.

-

Author Response

First of all, thank you very much for taking time out of your busy schedule to review my paper. Next I will answer your comments one by one.

Reply to reviewer 1:

Reviewers greatly appreciate the efforts that have been made by the author to improve the quality of their articles after peer review. l reread the author's manuscript and further reviewed the changes made along with the responses from previous reviewers' comments Unfortunately. The authors failed to make some of the substantial improvements they should have made making this article not of decent quality with biased, not cutting-edge updates on the research topic outlined.In addition, the author also failed to address the previous reviewer's comments, especially on comments number 1 (lack of novel), 2 (not incorporate the literature), and 7 (not a comprehensive discussion) Thank you very much for the opportunity to read the author's current work。

A: I have added some references from recent years as required, and reorganized and rewritten the conclusion.

Reviewer 3 Report

No comments

Author Response

Reply to reviewer 3:

Thank you very much for your valuable time and effort in reviewing my thesis. Your professional knowledge and careful review comments are of great value to my research work.

Under your guidance, I have a deeper knowledge and understanding of my thesis. I have benefited greatly from your suggestions on the content, structure and language of the article. Through your review comments, I found some existing problems, and on this basis, I made corrections and improvements to improve the quality of the paper.

Your review process is rigorous and comprehensive, giving me full affirmation and encouragement, which makes me more firm in my research direction. Your recognition and support for my research work will undoubtedly be the most powerful driving force for me to move forward.

Thank you again for your careful review and valuable suggestions. I will incorporate your opinions into my research and continue to grow and make progress in my future academic exploration.

Sincerely wish you good health and success in your work!